# Child Maltreatment Types by Age: Implications for Prevention

**DOI:** 10.3390/ijerph21010020

**Published:** 2023-12-22

**Authors:** Kate Guastaferro, Stacey L. Shipe

**Affiliations:** 1Department of Social and Behavioral Sciences, School of Global Public Health, New York University, 708 Broadway, 6th Floor, New York, NY 10003, USA; 2School of Social Work, Binghamton University, 4400 Vestal Pkwy E, Binghamton, NY 13902, USA; sshipe@binghamton.edu

**Keywords:** prevalence, child maltreatment, prevention

## Abstract

Child maltreatment is a global public health issue known to affect an average of 600,000 U.S. children of all ages (0–18 years old) annually. However, a preponderance of preventive programs target children on the younger end of the spectrum, specifically those aged 0–5. Annual reports of the prevalence of maltreatment provide opportunities to analyze trends, but in 2009, these reports stopped reporting the ages of victims for each type of maltreatment (i.e., neglect, physical abuse, emotional abuse, and sexual abuse). This omission limits the ability to match (or design) prevention programs responsive to the ages of those at greatest risk. Using data from the National Child Abuse and Neglect Data System (NCANDS) from 2011–2020, this secondary data analysis describes trends for four types of maltreatment by age from an epidemiological perspective. Implications for practice (i.e., prevention) and policy are presented. The findings of this study offer the first step in what is hoped to be a line of research that seeks to identify, match, and/or develop evidence-based programs to prevent child maltreatment among the populations at highest risk.

## 1. Introduction

Child maltreatment (i.e., neglect, physical abuse, psychological abuse, and sexual abuse) is a public health issue of considerable magnitude. Associated with lifelong negative consequences, such as poor mental health, substance misuse, risky sexual behavior, chronic poor health, and delinquency [1,2,3], the estimated economic burden of child maltreatment exceeds USD 428 billion [4]. In the U.S., nearly 4 million referrals are made annually to child-protection service agencies indicating concern about the safety and well-being of children under 18, of whom >600,000 are determined to be victims [5]. Given the scope and economic burden, the prevention of maltreatment is an urgent priority.

Over the last 20 years, the prevention of child maltreatment adopted a public health approach [6], transitioning from solely reacting to instances of maltreatment to working to prevent them from occurring in the first place. One of the most common preventive approaches to child maltreatment is home visiting, a service-delivery strategy in which the parents of young children are paired with a support person (provider) who visits the family’s home (or location of choice). Home visiting programs focus on the first five years of life and develop parental knowledge and skills related to child health and well-being, child development and school readiness, parent–child relationships, and family functioning. However, child maltreatment is a phenomenon that spans the developmental spectrum (0 to 18). Indeed, recent national data indicate that 28% of victims were under the age of 2, 17% were between the ages of 3 and 5, 25% were between 6 and 10, 19% were between 11 and 14, and 10% were 15 and older [5]. As maltreatment affects all ages under 18, to influence public health, evidence-based prevention programs should correspond to the age of the at-risk child.

In the U.S., the annual prevalence estimates of child maltreatment are collected by the National Child Abuse and Neglect Data System (NCANDS) and presented in annual reports produced by the Children’s Bureau. The NCANDS was established in 1988, and its data provide the opportunity to analyze important trends (e.g., the ages of victims, perpetrator type, and reporter type) over time that ultimately shape practice and policy. Prior to 2009, the NCANDS reported the proportion of victims by age for each type of maltreatment (e.g., medical neglect, neglect, other abuse, physical abuse, psychological abuse, or sexual abuse). By no longer reporting the prevalence of the sub-types of maltreatment by age, the NCANDS limited the ability to match (or design) prevention programs tailored to the developmental stage of the population at greatest risk. Given the stable prevalence of victims of child maltreatment in the U.S., it is important to critically examine the way in which prevention efforts align with those identified as being at risk. This study sought to fill an important gap in the literature on child-maltreatment prevention by presenting trends from an epidemiological perspective for four types of maltreatment by age, using nationally reported data on substantiated cases of maltreatment. The findings will be useful to and inform future prevention strategies.

## 2. Materials and Methods

We used the NCANDS Child File from 2011–2020, provided by the National Data Archive on Child Abuse and Neglect (NDACAN) at Cornell University. The years 2011–2020 were selected to create a comprehensive 10-year review of trends. The NCANDS data record case-level information on reports of child abuse and neglect from all 50 states, the District of Columbia, and Puerto Rico. The Child File includes case-level data for each report of alleged maltreatment that rises to the attention of child-protection services in the form of an investigation or alternative response.

The Children’s Bureau reports analyze allegations attached to individual children [5], which limits understanding as to how many allegations actually enter the system and answers the question of how many different types of maltreatment children experience. However, we were interested in understanding the number of occurrences of each type of maltreatment, regardless of the individual child. Therefore, we used all allegations over the 10 years of available records (*N* = 27,599,672). The NCANDS data record multiple allegation types per victim using the following codes (defined by the reporting states and territories): physical abuse, neglect or deprivation of necessities, medical neglect, sexual abuse, psychological or emotional maltreatment, sex trafficking, no alleged maltreatment, or other. For this analysis, we combined neglect and medical neglect, as well as sexual abuse and sex trafficking. We opted to exclude ‘other’ and ‘no alleged maltreatment’ types because there was no conceptual frame through which to make a decision about the meaning of these allegations. As a result, a total of four maltreatment types were included in the analysis: (1) physical abuse; (2) neglect; (3) sexual abuse; and (4) psychological maltreatment.

Each individual allegation in the Child Files was coded as either 0 (no), or 1 (yes). For example, if a record had a physical abuse and neglect allegation, then the data were coded 1 for physical abuse and 1 for neglect, but 0 for sexual abuse and 0 for psychological maltreatment. Once coded, the Child Files (*N* = 10) were merged into a master dataset. The analyses were descriptive in nature and focused on the prevalence of each maltreatment type by age group. To maximize visualization, we plotted by type of maltreatment and age group. All analyses used Stata Version 17 [7].

## 3. Results

The total numbers of allegations for the four types of maltreatment are presented in Table 1. Figure 1 depicts the prevalence of the four types of maltreatment by age group between the years of 2011 and 2020. Figure 1A depicts the prevalence of physical abuse, and the trend lines are fairly consistent for each age group. Over the nine years of the study, the prevalence of physical abuse among the 0-to-4 age group ranged from 29% to 27%, consistently reflecting the lowest prevalence among the groups. The prevalence in the 12to-17 age group similarly ranged between 29% and 32%. The greatest prevalence was consistently observed among the 5-to-11 age group, ranging from 41% to 43%.

The prevalence of neglect between 2011 and 2020 is depicted in Figure 1B. The lowest prevalence was consistently observed among the 12-to-17 age group, ranging from 23% to 27%, with an increasing trend throughout the years for which data were available. In 2011, the prevalence of neglect was approximately the same between the 0-to-4 and 5-to-11 age groups (38% and 39%, respectively). Over time, the 0-to-4 group had an observable decreasing trend, ranging from 38% to 34%, whereas the prevalence among the 5-to-11 group was fairly stable, ranging from 39% to 41%.

Figure 1C depicts the prevalence of sexual abuse from 2011 to 2020. The lowest prevalence was observed among the 0-to-4 age group, with a noticeable declining trend over time, ranging from 23% to 17%. Between 2011 and 2017, the highest prevalence was observed among the 5-to-11 age group, ranging from 42% to 43%. In 2018, a declining trend in prevalence (42–40%) among the 5-to-11 age group was contrasted with an increasing trend among the 12-to-17 age group (41–43%), moving the older age group to the highest prevalence designation. The prevalence of sexual abuse among the 12-to-17 age group ranged from 36% to 43%, with the notable exception of 2016, when the prevalence dropped to 31%.

The prevalence of psychological abuse from 2011 to 2020 is depicted in Figure 1D. The 5-to-11 age group consistently had the highest observed prevalence among all the age groups, ranging from 40% to 42%. The lowest prevalence was observed among the 12-to-17 age group, from 2011 to 2013 (29%), but switched to the 0-to-4 age group in 2014 (27–25%). The prevalence among the 12-to-17 age group steadily increased between 2014 and 2020, ranging from 31% to 34%.

Figure 2 compares the prevalence for the four types of maltreatment by age group from 2011 through 2020. The type of maltreatment with the consistently highest observed prevalence for the 0-to-4 age group was neglect, and the type of maltreatment with the lowest prevalence was sexual abuse (Figure 2A). The prevalence of physical and psychological abuse for the 0-to-4 age group was fairly even and consistent over time. For the 5-to-11 age group, the type of maltreatment with the lowest prevalence was consistently neglect (Figure 2B). In contrast, for this age group, the prevalence rates for physical, sexual, and psychological abuse fluctuated across the years of data analyzed. For the 12-to-17 age group, sexual abuse was consistently the type of maltreatment with the highest prevalence, including the anomaly of 2016 (Figure 2C). The least prevalent type of maltreatment observed for 12-to-17-year-old adolescents was consistently neglect. The rates of physical and psychological abuse were fairly consistent over time.

## 4. Discussion

To say the prevalence of any type of maltreatment is low is an oxymoron—any report of maltreatment is one too many. However, to prevent maltreatment from occurring, it is important to better understand the landscape of experiences across all developmental stages. The present analysis indicates that the prevalence of different forms of maltreatment varies across the developmental spectrum (0–17). Across the four subtypes of maltreatment included in the present analysis, the 5-to-11 age group consistently had the highest prevalence rates between 2011 and 2020. Within this group, physical abuse was the most prevalent form of maltreatment, whereas neglect was the lowest. The fact that the prevalence is highest among the 5-to-11 age group is not entirely surprising—the beginning of this age group is when children enter formal school environments and have contact with teachers. The consistent contact with students over time make teachers (and schools) potent points of intervention [8], but also increases surveillance and, thus, reports of maltreatment. The lowest rates of neglect were observed among the 12-to-17 age group; however, this must be interpreted with caution, as the definition of neglect (i.e., failure to meet basic needs) may not be applied to an adolescent population whose members are able to provide a level of basic needs for themselves [9], as in younger age groups (i.e., who are more dependent). Within the 12-to-17 age group, the highest prevalence was observed for reports of sexual abuse. However, overall, the age group with the highest prevalence rate of sexual abuse across the 9 years of data was the 5-to-11 age group, which was consistent with prior research [10]. The lowest prevalence for sexual abuse and psychological abuse was observed among the 0-to-4 age group, which makes sense, as these forms of maltreatment often require disclosure from the victim directly. Within the 0-to-4 age group, physical abuse had the highest prevalence, which aligns with the focus of prevention efforts on parenting behaviors to reduce problem behaviors [11].

A sufficient, yet effective, prevention program must be tailored to accommodate the heterogeneity in the prevalence of different forms of maltreatment across the developmental spectrum. The findings of this study offer the first step in a larger line of research that seeks to identify, match, and/or develop evidence-based programs to prevent child maltreatment among the populations at the highest risk.

### 4.1. Implications for Prevention

We see this study as the first step in understanding the ages and types of maltreatment that programs should be designed to target. Waid and Choy-Brown [12] defined primary prevention programs as those that limit initial exposure to maltreatment risk, secondary programs as those designed to reduce identified risk factors prior to the occurrence of maltreatment, and tertiary programs are those that limit the sequelae of maltreatment and aim to prevent recurring events. Jones-Harden et al. [13] suggest that these points of intervention are best paired with specific prevention strategies: primary prevention efforts typically adopt a universal approach (i.e., geared to the whole population or vulnerable sub-populations with no evidence of maltreatment); secondary efforts are selective strategies (i.e., they reduce harm among individuals at elevated risk); and tertiary efforts are indicated preventive interventions (i.e., aimed at reducing recurrence, or to mitigate adverse outcomes).

The majority of prevention programs have targeted parents through a selective or indicated approach as, most often, maltreatment is a product of parents’ acts of commission or omission. However, parenting is but one part of prevention [14]. Schools offer a complementary venue for universal prevention education, as evidenced by their use of programs focused on alcohol and substance abuse [15], mental health [16], and bullying [17]. To date, the majority of school-based child-maltreatment programs have focused on sexual abuse [18]. An exception is Childhelp’s Speak Up Be Safe, a universal program delivered through 12th-grade classrooms focused on all forms of maltreatment, including neglect and online bullying [19]. Our findings indicate that students aged 5 to 11 years old would benefit from universal preventive education programs covering all types of maltreatment. This is not to claim that the burden to protect themselves should fall on children alone; rather, we mean to suggest that to prevent all forms of maltreatment, children are important pieces in the prevention puzzle.

### 4.2. Implications for Policy

Federal initiatives in the U. S. have invested considerably in child-maltreatment prevention, beginning, in 1974, with the Child Abuse Prevention and Treatment Act (CAPTA; P.L. 93–247). In 2010, the Affordable Care Act initiated the Maternal Infant Early Childhood Home Visiting (MIECHV) program, administered by the Health Resources Services Administration allocating USD 1.5 billion specifically to home-visiting services. The MIECHV was reauthorized in 2022, with federal support increasing to USD 500 million annually through 2027. The Family First Prevention Services Act of 2018 (P.L. 115–123) amended programs in Title IV-B and Title IV-E of the Social Security Act to prioritize the prevention of child maltreatment, increase family preservation, and promote permanency for those in out-of-home care. The expansion of funding provided by Family First and MIECHV created a unique opportunity for child-maltreatment-prevention programs. However, the stewards of these funds must ensure prevention investments—primary, secondary and tertiary—are equitably distributed across all ages at risk of maltreatment.

### 4.3. Implications for Research

There is a need for clear investment in research to support the development and/or evaluation of preventive education programs targeting developmental ages >5 years. Of course, there are programs targeting parents of children over age 5, of which some are now supported by evidence, including Childhelp’s Speak Up Be Safe [19], but no programs targeting children over age 5 have been designated as ‘well-supported by research evidence’ by clearinghouses, such as the California Evidence-Based Clearinghouse (https://www.cebc4cw.org/ (accessed on 15 December 2023)). Research is needed to enhance the evidence of existing programs and, as has been indicated, future research must improve the effectiveness and implementability of these programs [20] to ensure they reach the targeted population.

### 4.4. Limitations

The findings presented here are informative, but should be interpreted with caution, as there are important limitations to the data. First, the data were limited to NCANDS, which reflects only cases that rose to the level of child-welfare involvement and were substantiated. The true prevalence of child maltreatment is likely to be much higher. Second, the data are limited to the years between 2011 and 2020. The comparability of the data for 2021 and 2022 is not known, as the prevalence of maltreatment was atypical during these years [21]. Third, it is well-acknowledged that there is a high rate of co-occurrence between different types of maltreatment [22,23]; for example, in a study of *N* = 303 9-to-12-year-old adolescents, child-protection-service reports indicated that neglect was most commonly accompanied by physical or emotional abuse. In this analysis, we dichotomized the experience of maltreatment, which may not capture the severity or complexity of maltreatment experiences. National data systems do not capture these nuances, limiting the field as a whole. The provision of federal resources devoted to programs is an important step for prevention, but investment in the infrastructure and requirements of reporting systems is also warranted. The present analysis was also limited to data from the U.S.; it may be prudent to examine the trends of other countries and the programs or policies in place to provide comprehensive preventive interventions.

## 5. Conclusions

The current paper was not designed to offer specific suggestions as to how to prevent different types of child maltreatment at different developmental stages. Instead, we view the current paper as a first step in a line of research aiming to increase the potency of child-maltreatment-prevention efforts. Child maltreatment is a public health problem affecting children between 0 and 18 years of age. However, the majority of federally supported prevention efforts focus only on children under 5. While other research has examined trends of substantiation by victim age over time [24], we extend this knowledge base by using nationally available prevalence data across 10 years to demonstrate that children of different age strata experience different types of maltreatment at different frequencies. To truly impact the prevalence rates of child maltreatment, prevention programs must encompass all forms of maltreatment across all developmental stages. The next step in this line of research is a careful mapping of evidence-based programs that demonstrate a reduction in risk for child maltreatment across developmental stages and a critical examination of the types of maltreatment these programs are found to affect.

## Figures and Tables

**Figure 1 ijerph-21-00020-f001:**
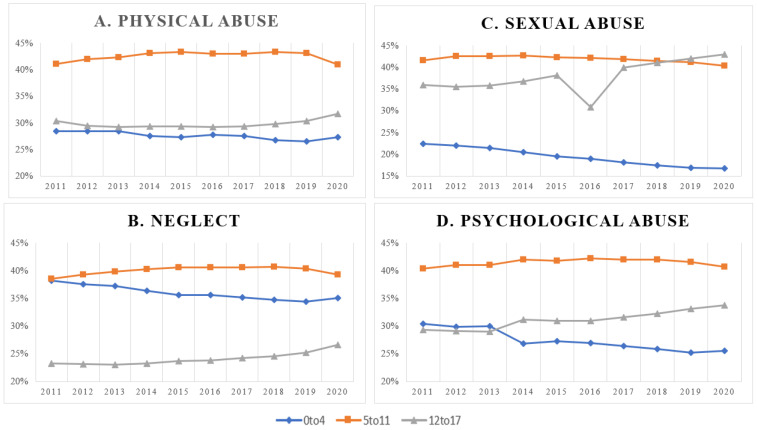
Prevalence of subtypes of maltreatment by age group between 2011 and 2020.

**Figure 2 ijerph-21-00020-f002:**
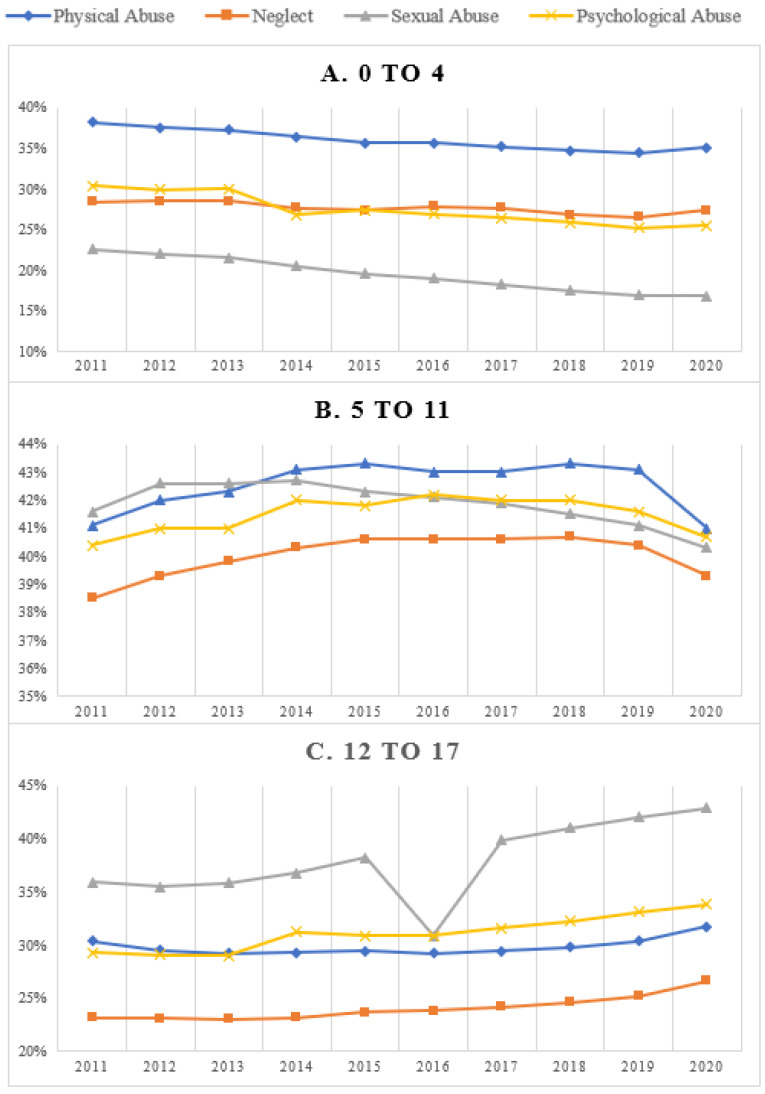
Prevalence of subtypes of maltreatment across 2011 and 2020 by age group.

**Table 1 ijerph-21-00020-t001:** Allegation types by year.

Year	Neglect	Physical Abuse	Sexual Abuse	Psychological Maltreatment	Total
2011	2,719,392	939,424	307,856	355,594	4,322,266
2012	2,250,666	796,233	263,469	282,174	3,592,542
2013	1,920,956	677,247	222,510	239,026	3,059,739
2014	1,714,509	589,280	189,058	187,901	2,680,748
2015	1,645,641	585,459	178,664	191,510	2,601,274
2016	1,542,898	596,411	174,279	181,084	2,494,672
2017	1,495,698	571,931	166,989	185,915	2,420,533
2018	1,413,308	535,929	167,345	184,952	2,301,534
2019	1,309,072	512,458	157,651	183,623	2,162,804
2020	1,187,440	425,232	139,250	165,856	1,917,778
**Total**	17,199,580	6,229,604	1,967,071	2,157,635	27,599,672

## Data Availability

Data are available from the National Data Archive on Child Abuse and Neglect (https://www.ndacan.acf.hhs.gov/).

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
