# Peer review of "Child Maltreatment Types by Age: Implications for Prevention"

_ijerph, 2023, doi:10.3390/ijerph21010020_

Round 1
Reviewer 1 Report
Comments and Suggestions for Authors
Title:
Child maltreatment types by age: Implications for prevention
The reviewer’s comments
The reviewer would like to see some revisions made to your manuscript.
1) The abstract will be revised to include following: Background, Context, Method, Data collection and conclusion. Therefore please modify the abstract. Thanks.
2) Increase the discussion of literature on child maltreatment types by age
3) In the section of the discussion, I suggest the authors should provide more theoritical literatures to dialogue with the results. Additional relevant studies should be included in order to enhance discussion.
4) Please strengthen the conclusion and implications. Good finding suggestions for future practitioners and researchers, and enhance the whole quality of this article.
5) Major revision.

Title:
Child maltreatment types by age: Implications for prevention
The reviewer’s comments
The reviewer would like to see some revisions made to your manuscript.
1) The abstract will be revised to include following: Background, Context, Method, Data collection and conclusion. Therefore please modify the abstract. Thanks.
2) Increase the discussion of literature on child maltreatment types by age
3) In the section of the discussion, I suggest the authors should provide more theoritical literatures to dialogue with the results. Additional relevant studies should be included in order to enhance discussion.
4) Please strengthen the conclusion and implications. Good finding suggestions for future practitioners and researchers, and enhance the whole quality of this article.
5) Major revision.
Reviewer 2 Report
Comments and Suggestions for Authors
Good afternoon,
Please find attached the report of the article.
Regards

Reviewer 3 Report
Comments and Suggestions for Authors
The authors need to specifically say why the 0-5 age group only has been focused by the government, and the other age groups heave been neglected by the government. And the authors need to say why the 2011-2020 data was selected, and why the other years data was not used.
Round 2
Reviewer 1 Report
Comments and Suggestions for Authors
tle:
Child maltreatment types by age: Implications for prevention
The reviewer’s comments
Thanks to the author for the correction. Revisions or explanations are all made according to the suggestions of the reviewers. Accept in present form.
Comments on the Quality of English Language
tle:
Child maltreatment types by age: Implications for prevention
The reviewer’s comments
Thanks to the author for the correction. Revisions or explanations are all made according to the suggestions of the reviewers. Accept in present form.
Reviewer 3 Report
Comments and Suggestions for Authors
None